# Anatomical Variations of the Common Carotid Arteries and Neck Structures of the New Zealand White Rabbit and Their Implications for the Development of Preclinical Extracranial Aneurysm Models

**DOI:** 10.3390/brainsci13020222

**Published:** 2023-01-28

**Authors:** Gwendoline Boillat, Tim Franssen, Stefan Wanderer, Jeannine Rey, Daniela Casoni, Lukas Andereggen, Serge Marbacher, Basil E. Gruter

**Affiliations:** 1Department of Neurosurgery, Kantonsspital Aarau, 5001 Aarau, Switzerland; 2Cerebrovascular Research Group, Department for BioMedical Research, University of Bern, 3010 Bern, Switzerland; 3Experimental Surgery Facility, Department for Biomedical Research, Faculty of Medicine, University of Bern, 3010 Bern, Switzerland; 4Institute of Neuroradiology, Department of Radiology, Kantonsspital Aarau, 5001 Aarau, Switzerland

**Keywords:** aneurysm, animal model, New Zeeland White Rabbit, carotid arteries, anatomy

## Abstract

Background: Rabbit models involving neck arteries are of growing importance for the development of preclinical aneurysm models. An optimal understanding of the anatomy is primordial to allow the conception of models while minimizing mortality and morbidity. The aim of this study is to give reliable anatomical landmarks to allow a standardized approach to the neck vessels. Methods: We performed a necropsy on nine specimens from ongoing experimental studies. We measured the distance between the origins of the right and left common carotid artery (rCCA/lCCA) and between the rCCA and the manubrium sterni (MS). The structures at risk were described. Results: Female New Zealand White rabbits (NZWR) weighing 3.7 ± 0.3 kg and aged 25 ± 5 weeks were included. The rCCA origin was located 9.6 ± 1.2 mm laterally and 10.1 ± 3.3 mm caudally to the MS. In all specimens, the lCCA originated from the aortic arch, together with the brachiocephalic trunk (BCT), and 6.2 ± 3.1 mm proximally to the rCCA origin. The external and internal jugular veins, trachea and laryngeal nerve were the main structures at risk. Conclusions: The data help to localize both CCAs and their origin to guide surgical approaches with the manubrium sterni as a main landmark. Special attention has to be paid to the trachea, jugular veins and laryngeal nerves.

## 1. Introduction

Due to its availability and easy handling, the rabbit model has been widely established and well characterized for preclinical studies [1]. Its vascular anatomical, physiological and hemodynamical characteristics, close to humans, make it an ideal model for vascular pathologies and, more specifically, to study the pathophysiology and treatments of aneurysms [2,3,4,5,6,7,8]. The surgical creation of aneurysms is usually performed on rabbit neck arteries, which have a comparable diameter to the human brain’s major arteries (such as the middle cerebral artery). Among the various existing models, the elastase-digested stump aneurysm, consisting of an endovascular application of the elastase enzyme at the origin of the right common carotid artery (rCCA) and the ligation of the artery, is probably the most common one [9,10,11,12]. However, the procedure can be associated with complications and additional mortality due to the presence of the aberrant origin of the tracheoesophageal branches and the superior thyroid artery from the proximal portion of the CCA [13]. In order to overcome this issue, research groups developed an open technique to expose the rCCA at its origin and surgically temporarily clip the vessel during the application of the elastase solution and ligate the artery afterward [9,12]. Furthermore, complex aneurysm creation, such as bifurcation aneurysm models, always implies an open approach, including the dissection of the neck arteries and microsurgical anastomosis [14,15,16], where a good knowledge of anatomy is mandatory to avoid lesions of vital structures, such as the trachea, great vessels (arteries and veins) and nerves of the neck, as well as the pleura. However, the exact vascular anatomy of the rabbit is not well described, and previous radiological studies have suggested a high variability in the anatomical position of vascular structures [17,18]. Moreover, the region of the CCA’s origin contains the above-mentioned vital structures, which can be at risk of damage during aneurysm creation surgeries.

The aims of this work are to describe reliable landmarks for the localization of the origin of the CCAs; to observe how the anatomical variation of the vessels may impact the surgical approach; and to describe which anatomical structures are at higher risk of damage during surgeries on the neck arteries.

## 2. Materials and Methods

### 2.1. Study Design and Animals

In this study, *n* = 9 cadavers of female New Zealand White (NZW) rabbits (Charles River Laboratories) weighing 3.7 kg (±360 g) and with a mean age of 25 ± 5 weeks were included. Specimens were received after euthanasia, performed in the setting of previous experimental series. The original studies were performed in accordance with the institutional guidelines for the care and use of experimental animals and following the ARRIVE guidelines. They were approved by the animal care committee of the Canton of Bern, Switzerland (approval number BE 108/16). Euthanasia was performed by intravenous injection of pentobarbital 120 mg/kg (Esconarkon ad us. vet., Streuli, Switzerland). Dissection and anatomical measures were performed immediately after euthanasia. 

### 2.2. Approach and Measures

Each rabbit was dissected in the region of the neck from the larynx to approximately the third rib in order to expose the great vessels of the neck and the surrounding structures. After a median skin incision, the musculature of the neck was exposed and dissected. The sternocephalicus muscle (SCM) was reclined rostrally in order to expose the underlying vessels. The tissues around the CCAs were dissected to allow good visibility of the vascular structures and surrounding nerves. To access the region of the brachiocephalic trunk and the rCCA origin, the medial part of the first rib was removed. The brachiocephalic trunk (BCT), rCCA and left CCA (lCCA) were exposed to allow for measurements. The distance between the rCCA origin and manubrium tip (incisura jugularis) was measured in craniocaudal and mediolateral directions using a tape ruler and a measuring clip. In the same way, the distance between both rCCA and lCCA origins was recorded. Lastly, we identified both CCAs’ origins and classified them into 3 variations, based on the previously described classification by Ding et al. [17]: Type 1: lCCA originating from the bifurcation of the aortic arch and the BCT; Type 2: lCCA originating from the aortic arch; Type 3: lCCA originating from the BCT itself, next to the rCCA (Figure 1).

### 2.3. Neck Dissection

In order to describe the relevant anatomical structures at risk during an approach to the rCCA, we performed a complete neck dissection. Each step was photographically documented and every structure was described, based on anatomical atlas references [19,20]. First, a median skin incision was performed from the manubrium tip to the hyoid bone. The fat pad was removed in order to expose the superficial musculature (Figure 2A). The descending pectoral muscle (DPM) (Figure 2B), first rib and tip of the manubrium were removed to allow access to the caudal part of the great vessels (Figure 2C). Meticulous soft tissue dissection around the vascular and neural structures was performed in order to follow the courses of the structures. Both sternocephalicus muscles (SCMs) were cranially reclined to access the deep musculature (Figure 2D), and finally, the sternothyroid muscle (STM), running above the trachea, was cranially reclined, and pericardial fat was exposed (Figure 2E). Pericardial fat was finally removed to expose the heart and the origin of the great vessels (Figure 2F). The structures identified to be at risk during the approach to the neck vessels were defined as structures in direct contact with the carotid arteries or structures that have to be strongly manipulated during the approach. These were documented in detail. 

## 3. Results

The tip of the manubrium sterni is a reliable landmark to guide dissection and find the origin of the rCCA, which should be located about 1 cm laterally and 1 cm caudally to it. The sternocephalicus muscle is a good landmark to find and follow the course of the CCAs on both sides of the trachea. During dissection, special attention should be paid to the jugular veins, the laryngeal nerves and the trachea itself.

### 3.1. Distance between rCCA’s Origin, Manubrium and lCCA’s Origin

The mean distance between the origin of the rCCA at the BCT and the tip of the manubrium in the mediolateral direction was 9.6 mm ±1.2 mm. In the craniocaudal direction, the mean distance was 10.1 mm ±3.3 mm. Detailed data are shown in Table 1 and Figure 3. The mean distance separating the origin of the rCCA and the lCCA was 6.2 mm ±3.1 mm. Detailed data are shown in Table 2 and Figure 3.

### 3.2. Variations of CCA’s Origin 

In all nine animals (100%), the rCCA originated from the BCT, and the lCCA originated from the bifurcation of the BCT and the aortic arch, corresponding to the Type 1 variation. No rabbits showed Types 2 or 3 vessel locations with lCCA directly originating from the aortic arch or lCCA originating next to the rCCA from the BCT. We also did not observe the aberrant variation of the subclavian artery (SCA) anatomy (Figure 1).

### 3.3. Descriptive Anatomy

After skin incision and subcutaneous fat tissue dissection, the first layer of muscles inserting on the manubrium sterni becomes visible: first, the sternocephalicus muscle (SCM) and then, medially to it and laying above the trachea, the sternothyroid (STM) and the sternohyoid muscles (SHMs). The transverse/descending pectoral muscles (TPM/DPM) find their origin at the manubrium sterni and insert laterally in the crista humeri (Figure 4A). The SCM covers the CCAs and the internal jugular veins (IJVs) on both sides of the trachea and has to be dissected and reclined to access the vessels. Cranially, the CCAs run medially to the SCM, whereas caudally, they run more laterally (Figure 4C). Under the SCM and laterally to the STM and SHM run the CCAs, the vagal nerve and the IJV. The three structures are bounded together with connective tissue. Cranially, these connective tissues are rather loose, and the distance between the artery and vein measures about 5–10 mm. However, caudally, the EJV comes medially in contact with the artery, and the nerve crosses in between. This makes a dissection more difficult in the caudal area (Figure 4D and Figure 5). In most cases, a transversal jugular vein can be seen at the level of the manubrium sterni and limits further dissection caudally. The vein is an anastomosis between both EJVs and can be cut in the middle to allow access to the caudal portion of the great vessels (Figure 4B). Further, caudally, the rCCA runs under the clavicle and comes above the trachea to join the rSCA at the BCT. Laterally, the vagal nerve (VN) crosses above the SCA and the EJV and IJV and meets the right subclavian vein (rSCV). The BCT meets the aortic arch and both aorta and SCV join the heart. The thymus can be observed superficially covering this part of the vessels. On the left side, the lCCA crosses above the trachea to join the origin of the BCT on the aortic arch. Figure 5 gives an overview of the deep neurovascular structures after the resection of the muscular and bony layers.

## 4. Discussion

The rabbit aneurysm model is of great interest for the study and better understanding of the pathophysiological aspects of human intracranial aneurysms [4,5,7,21,22,23,24,25,26,27]. Scientists are constantly developing new models or refining the existing models for the creation of aneurysms on rabbit neck arteries [5,6,7,9,11,14,16,28,29,30,31,32]. The elastase-induced aneurysm model is one of the most commonly used and well described [28,29,33,34,35,36,37] and is widely used for endovascular device testing [38,39,40,41,42,43,44,45,46,47,48]. However, in a previous series, the periprocedural mortality of the rabbit elastase model was reported at around 8% [49,50]. Given the number of studies that conceal complications and mortality rates, this number could be even higher. For instance, research investigating the morbidity of the model showed complication rates reaching >50% [49,50,51,52]. As morbidity and mortality could influence the quality of the scientific result [53], and given the growing importance of animal welfare in preclinical studies, the impact of complications must not be underestimated and should be a concern for every scientist. 

If the age and weight of the animal, as well as the duration of the procedure, have been demonstrated to be determinizing factors for postoperative complications, the direct injury of the structures of the neck through endovascular or surgical procedures are recognized to be a critical source of failure. In order to efficiently improve the techniques, avoid pitfalls and develop new approaches with lower morbidity and mortality, the clinician and/or scientist have to acquire sufficient knowledge of the anatomy of the animal. Unfortunately, the current literature and even the veterinary educational resources lack exact information about detailed anatomy and its possible variations, particularly in the region of the neck. This study gives anatomical reliable landmarks to allow a standardized approach to the neck vessels, with low morbidity and mortality. This study shows that in order to expose the origin of the rCCA, the manubrium sterni can be used to guide the dissection, which should be performed about 1 cm laterally and caudally to the incisura jugularis. There are no relevant variations to expect between animals of the same breed and with similar ages, as we found only variability of a few millimeters between the rabbits. However, the distance between the rCCA and lCCA origins varied between 4 and 12 mm in this study. Thus, the origin of one CCA should not be used to guide the dissection of the second one, and dissection from cranial to caudal is recommended to find the origin of the CCAs. Instead, the SCM can be used to find and follow the CCAs, which run laterally to the caudal part of the muscle and medially to its cranial part. As the muscle plays a key role in head motion, especially rotation and inclination, lesions have to be avoided during dissection, and smooth instruments such as anatomical forceps and vessel loops are of great help for this step of the surgery. Furthermore, the SCM runs laterally to the external jugular vein (EJV), which is at risk of injury while dissecting the lateral part of the muscle. A wet swab provides good protection against sharp instrument manipulation and prevents dryness and weakening of the wall. 

On the other hand, Ding et al. already focused on the possible variations concerning the carotid arteries that may impact endovascular approaches in the elastase model [17]. The authors found three main variations of the carotid origins (Figure 1) that can impact catheterizing as well as open surgery. In the present study, both CCAs originate from the BCT according to the Type 1 variation described by Ding et al., with the rCCA originating from the BCT and the lCCA originating from the bifurcation of the BCT and aortic arch. This is also consistent with most of the data shown in the anatomical atlas references [19,20]. Variations of these origins or other uncommon variant arteries may, however, significantly affect the surgery and even impact the testing of endovascular techniques in an aneurysm model. Although rare, these variations have, thus, to be known and taken into consideration while working with such models. Furthermore, some authors experimented with dramatic tracheal necrosis and hemorrhage after endovascular elastase application in order to create an aneurysm [13,52,54], revealing the presence of aberrant superior thyroid arteries or anastomosis from the carotid arteries. This complication can be avoided by proceeding with open procedures, which allows the closure of the variant branches [9,12,33,55,56]. In order to perform safe interventions, anatomical landmarks and specific anatomical descriptions are definitely needed. Surgeons performing an approach to the great neck vessels have to pay special attention to numerous vital structures. The jugular veins, which run directly laterally to the rCCA and cross the right subclavian artery to form the rSCV, are at high risk of damage during the dissection of the proximal part of the CCAs. At the level of the rCCA origin from the BCT, the EJV comes directly in contact with the artery and may be adherent with it. The dissection has to be performed carefully on the side of the artery in order to avoid any damage to the venous wall. Such injuries are often untreatable and may result in the death of the animal. A different, important structure is the vagal nerve, including its laryngeal branches, which run directly with the CCAs and sometimes form the nervous plexus, which can be easily damaged by dissecting the vessels. Such injuries may lead to laryngeal paresis, which appears clinically as postoperative stridor and increases inspiratory effort. According to the severity of the damage, this can elevate intrapulmonary negative pressure with consecutive pulmonary edema, respiratory depression and death. Moreover, as illustrated in our study, the CCAs run along both sides of the trachea and cross above it to join the BCT and aortic arch. The trachea is a rather strong structure. However, direct pressure or traction during the dissection should be avoided in order to ensure correct ventilation during the surgery, and the surgeon should pay attention not to injure the tracheal wall using sharp instruments during the approach. Lastly, the thymus lays just above the proximal rCCA and BCT and may obstruct the dissection of the proximal part of the CCA. So far, injury of the thymus has not been described as a mortal lesion, and it does not seem to cause any relevant morbidities in the postoperative phase. However, swelling and bleeding due to rough manipulation can significantly complicate the dissection, and we recommend preserving the gland as long as possible with the application of a wet swab.

The presented comprehensive overview should help surgeons to plan their operation and to minimize perioperative morbidity and mortality. In the same way, it is supposed to improve scientific results and support the development of further, sophisticated aneurysm models.

## 5. Conclusions

Our study demonstrates the tip of the manubrium sterni being a reliable landmark to guide dissection and find the origin of the rCCA, which should be located about 1 cm laterally and 1 cm caudally to it. The sternocephalicus muscle also helps to find and follow the course of the CCAs on both sides. Great variations of the CCAs’ origin are rare within the NZW rabbit strain, and the insights into the specific anatomy of the neck provided in this study should help surgeons to avoid complications and improve surgical and scientific results.

## Figures and Tables

**Figure 1 brainsci-13-00222-f001:**
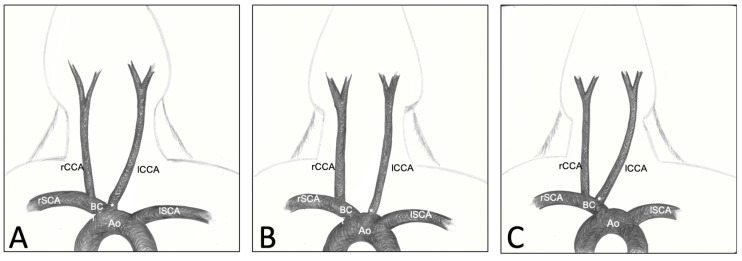
Classification of the three CCAs’ origin variations. Type 1: lCCA originating from the bifurcation of the aortic arch and the BCT (**A**); Type 2: lCCA originating from the aortic arch (**B**); Type 3: lCCA originating from the BCT, next to the rCCA (**C**).

**Figure 2 brainsci-13-00222-f002:**
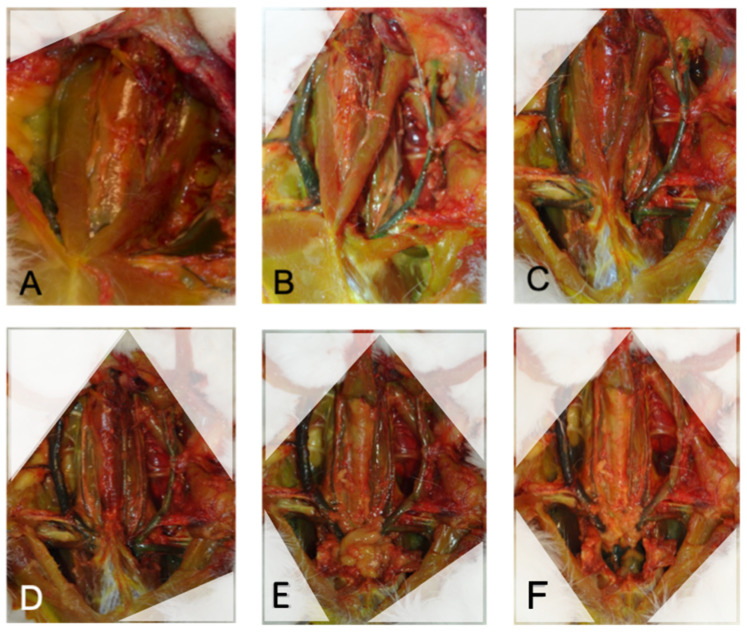
Illustration of the dissection step by step. (**A**) After median skin incision and fat pad resection. (**B**) After removal of the lDPM and first rib on the left side. (**C**) After removal of the rDPM and first rib on the right side. (**D**) After section and reclination of both SCM. (**E**) After section and reclination of the STM. (**F**) After dissection of the heart cavity. Note the reclined SCM/STM in the white triangles in (**D**–**F**).

**Figure 3 brainsci-13-00222-f003:**
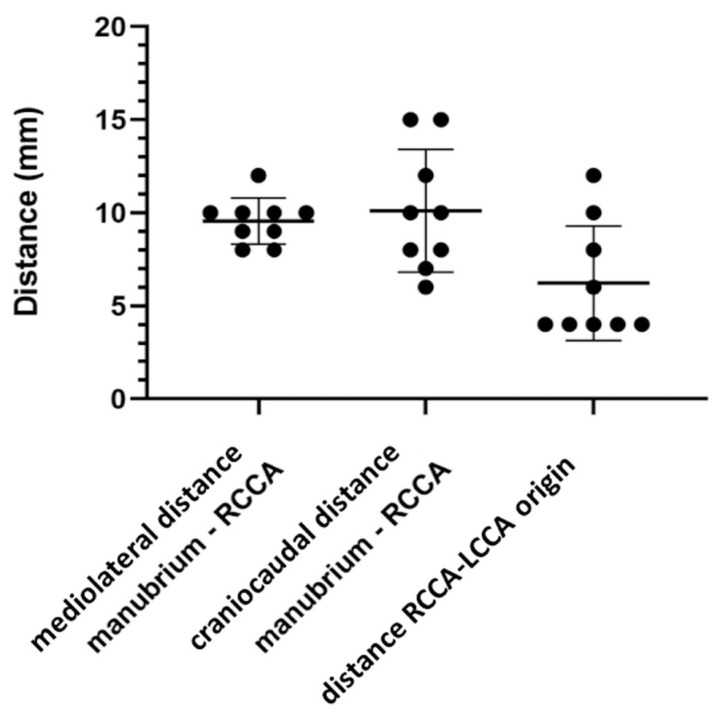
Measured distance between the tip of the manubrium and the origin of the right common carotid artery at the brachiocephalic trunk in a mediolateral direction (SD ± 1.2 mm), in the craniocaudal direction (SD ± 3.3 mm) and distance between RCCA and LCCA origins (SD ± 3.1 mm).

**Figure 4 brainsci-13-00222-f004:**
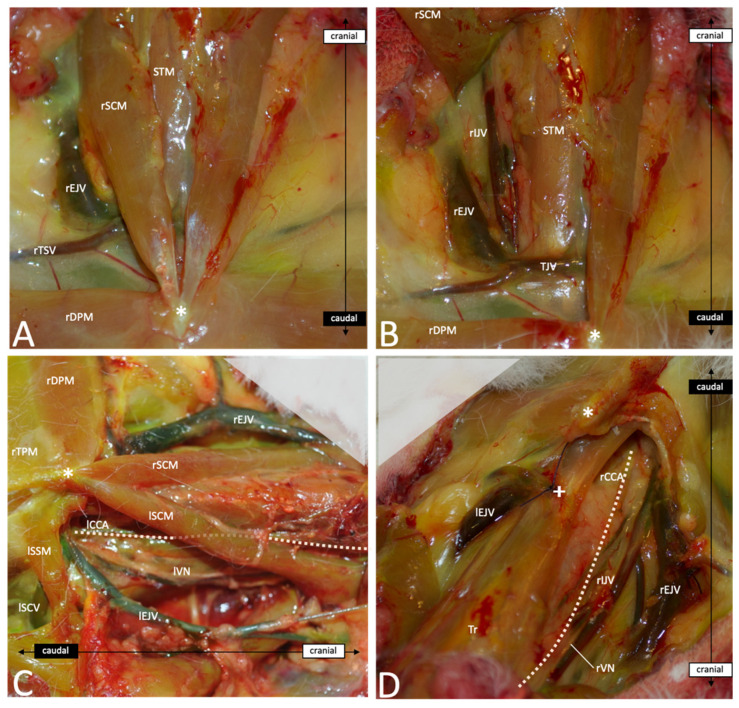
(**A**) Frontal view of the first superficial layer after skin incision and subcutaneous fat tissue dissection. (**B**) Frontal view after proximal section and reclination of the rSCM on the right side. (**C**) Left lateral view (surgical view for the approach on the left side of the neck) after resection of the lDMP and lTPM and dissection of the connective and fat tissue to free the neurovascular structures. The dotted line highlights the course of lCCA. (**D**) Cranial oblique overview on the right side (surgical view for the approach on the rCCA) after resection of the SCMs, the rDPM and rTPM and section of the TJV. The dotted line highlights the course of the rCCA. Abbreviations: *—tip of the manubrium sterni; +—section point of the transversal jugular vein; Tr—trachea; rSCM—right sternocephalicus muscle; lSCM—left sternocephalicus muscle; STM—sternothyroid muscle; rDPM—right descending pectoral muscle; rTPM—right transverse pectoral muscle; lSSM—left sternoscapular muscle; rEJV—right external jugular vein; lEJV—left external jugular vein; rIJV—right internal jugular vein; lIJV—left internal jugular vein; rTSV—right transverse scapular vein; TJV—transversal jugular vein; lSCV—left subclavian vein; lCCA—left common carotid artery; rCCA—right common carotid artery; lVN—left vagal nerve; rVN—right vagal nerve.

**Figure 5 brainsci-13-00222-f005:**
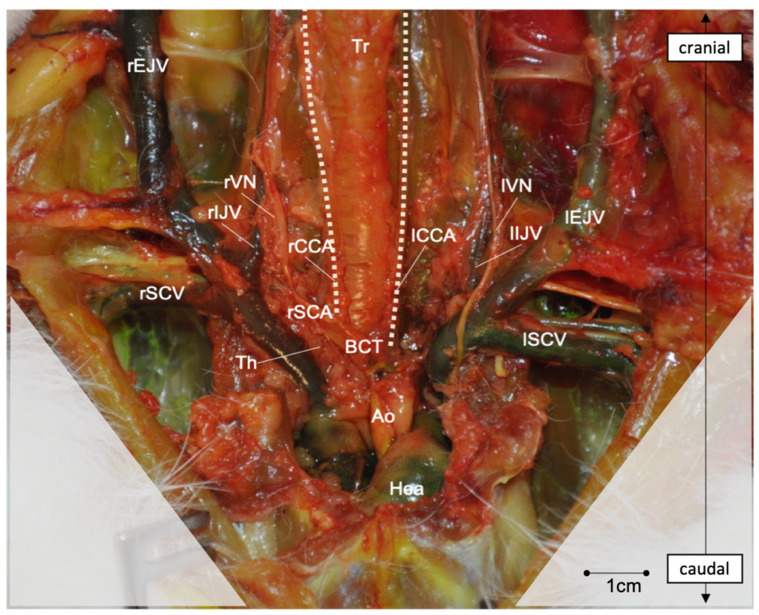
Frontal overview of the deep neurovascular structures of the neck after resection of the muscle layers and cranial part of the thoracic cage. SCM, STM, SHM, DPM, TPM, and SSM, as well as the clavicle, the three first ribs and the cranial third of manubrium sterni were removed. Subcutaneous fat, thymus and pericardial fat were dissected and removed to free the vessels and nerves. The dotted lines highlight the course of the rCCA and lCCA. Abbreviations: Tr—trachea; Hea—heart; Th—thymus; SCM—sternocephalicus muscle; STM—sternothyroid muscle; DPM—descending pectoral muscle; TPM—transverse pectoral muscle; SSM—sternoscapular muscle; rEJV—right external jugular vein; lEJV—left external jugular vein; rIJV—right internal jugular vein; lIJV—left internal jugular vein; rSCV—right subclavian vein; lSCV—left subclavian vein; Ao—aorta; BCT—brachiocephalic trunk; lCCA—left common carotid artery; rCCA—right common carotid artery; lVN—left vagal nerve; rVN—right vagal nerve.

**Table 1 brainsci-13-00222-t001:** Measured distance between the tip of the manubrium and the origin of the right common carotid artery at the brachiocephalic trunk in a mediolateral direction (SD ± 1.2 mm) and a craniocaudal direction (SD ± 3.3 mm).

Animal Number	Mediolateral Direction (mm)	Craniocaudal Direction (mm)
**1**	10	8
**2**	10	7
**3**	10	10
**4**	10	8
**5**	8	6
**6**	12	15
**7**	9	12
**8**	9	15
**9**	8	10
Mean (±SD)	9.6 (±1.2)	10.1 (±3.3)

**Table 2 brainsci-13-00222-t002:** Measured distance between both origins of the common carotid arteries (SD ± 3.1 mm).

Animal Number	Distance (mm)
**1**	4
**2**	4
**3**	6
**4**	4
**5**	12
**6**	4
**7**	8
**8**	4
**9**	10
Mean (±SD)	6.2 (±3.1)

## Data Availability

The data presented in this study are available on request from the corresponding author.

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
