# Peer review of "Anatomical Variations of the Common Carotid Arteries and Neck Structures of the New Zealand White Rabbit and Their Implications for the Development of Preclinical Extracranial Aneurysm Models"

_brainsci, 2023, doi:10.3390/brainsci13020222_

Round 1
Reviewer 1 Report
Comments and Suggestions for Authors
The authors described the anatomical variations of the common carotid arteries in rabbit model in order to improve the knowledge and support further development of aneurysm models. While the descriptions are of good quality the mentioned tables (Table 1 and Table 2) are missing from the manuscript. It would be relevant to insert tables containing detailed measurements for each specimen. Also, given the title of the manuscript, the authors should emphasize more in the discussion section about the implications of their results in the development of aneurysm models. Moreover, please make sure to check the spelling (e.g. page 2, rows 72 and 77, "rip" is used instead of "rib"; page 7, row 195, "structures: The", ":" can be replaced by "."). If available a supplementary video with the dissection steps would be a nice addition to the paper.
In conclusion, the authors need to address these issues, in order to consider the paper acceptable for publishing.
Author Response
Please see the attachement

Reviewer 2 Report
Comments and Suggestions for Authors
Boillat et al. presented an interesting manuscript entitled Anatomical variations of the common carotid arteries and neck structures of the New Zeeland White Rabbit and its implication for the development of preclinical extracranial aneurysm models. The topic is interesting and clinically useful. However, it contains errors by which. I have included my comments below:
- use the method with liquid stained plastic which will make the course of the analyzed vessels visible; it is difficult to see the described vessels without this. This is standard in angiological research.
- only one (middle) small paragraph has the character of a discussion; the rest is like results or some thoughts of the authors not supported by citations from the literature; this part needs a thorough rewriting
- numbers in abstract and keywords are unnecessary
- 20 citations for a full length article seems insufficient
- section 2.3 should be described in more detail
Author Response
Please see the attachement.

Round 2
Reviewer 2 Report
Comments and Suggestions for Authors
The manuscript has been significantly improved. I understand the Authors' explanation regarding the lack of use of the method with liquid-stained plastic. Despite this conceptual mistake in the planning of the study, I think the paper contributes valuable, very useful information. In my opinion, it should be published in this journal.